# What Is a Disease for Doctors? A Realist Thematic Qualitative Analysis of the Interpretation of Clinical Vignettes

**DOI:** 10.3390/healthcare12121228

**Published:** 2024-06-20

**Authors:** Fabrizio Consorti, Rossella Melcarne, Domenico Pisanelli, Chiara Scorziello, Laura Giacomelli

**Affiliations:** 1Department of General Surgery, University Sapienza of Rome, 00185 Rome, Italy; rossella.melcarne@uniroma1.it (R.M.); chiara.scorziello@uniroma1.it (C.S.); laura.giacomelli@uniroma1.it (L.G.); 2Institute of Sciences and Technologies of Cognition, National Research Council, 00196 Roma, Italy; domenico.pisanelli@cnr.it

**Keywords:** disease, philosophy of medicine, thematic analysis, clinical vignettes, critical realism, Italy

## Abstract

Given the long-standing debate about the nature of the concept of disease, the objective of this study was to understand how doctors categorize a condition as a disease or not, and what the kind of information they use is. A survey with a set of eighteen clinical vignettes was designed, and nineteen physicians and senior students purposefully selected were asked to interpret those situations as diseases or not and to produce an anonymous short written piece of text providing the motivation of their choice. Realist thematic analysis was used to analyse the answers, and four themes emerged: the temporal dimension of a disease, reification of disease, disease as an existential condition, and disease as a motivation to action. The respondents’ interpretations were very heterogeneous, supporting the idea that physicians do not share a common prototypical concept of disease. The results suggested that the interpretation of a condition as a disease or not is the final outcome of a process, in which information from objective, subjective, and socially mediated elements is taken into consideration. According to a critical realist and systemic approach, we hypothesize that the context of doctor–patient relationship could influence the interpretation of the same condition as being a disease or not.

## 1. Introduction

The attempt to conceptualize what a disease is has been a long matter of discussion, past and present [1]. To philosophers, the definitions of a disease has been historically divided into three main categories: pure naturalist [2], pure normative [3], and a pragmatic approach, representing a combination of the two. The naturalist definition considers disease as a biological change or dysfunction. In contrast, the normative definition is shaped by social norms, categorizing certain behaviours as undesirable or deviating from accepted moral and behavioural standards, out of the range of what is considered normal. The pragmatic definition considers a collection of biological, functional, and normative components [4,5,6].

Some philosophers have indeed questioned the need to define a disease, because “the concept of disease does not in fact play the crucial role in clinical decision making” ([7], p. 2). Despite his provocative and sceptical way of thought, Hesslow [7] listed four purposes for the definition of a disease: to allow the initiation of a diagnostic inquiry or medical treatment, to ensure welfare, to relieve one from the obligation to work, and, in the case of mental illness, to relieve moral responsibility and legal liability.

Others proposed a matter-of-fact approach to the problem, “a more realistic description of the way we build, use, apply, and change our concept of disease.” ([8], p. 424) and that “‘Disease’ is not a theoretical concept, but a practical term.” ([8], p. 432). The issue of adequately addressing such a concept becomes more complex once related but not overlapping concepts such as illness, sickness, dysfunction, disorder, and health are brought into consideration [9]. Diving deeper into the subject, there are voices not yet considered when attempting to define a disease: what do doctors and medical educators think? How do doctors and medical educators use the concept of disease in their professional decision-making and actions?

To the authors’ knowledge, only one study investigated whether doctors had a conceptual prototype of diseases. Hoffman conducted a quantitative study based on a survey of two hundred and twenty-three Norwegian physicians. The physicians were given an extensive list of conditions and were asked to rank them. Their rankings were based on the assumption that those conditions in the list were diseases. Because of the wide variability in response, the study falsified the hypothesis that disease was a prototypical concept. Unfortunately, because of its quantitative approach, this study did not explain the variability observed in labelling a condition as a disease or not [10].

From an educational perspective, the only apparent concern is to support students in developing their concept of illness as a dimension of patient suffering, with the same importance as the concept of disease as an organic or functional alteration [11,12]. This could be achieved through developing communication skills and understanding of the patient’s background not only on a pathophysiological level, allowing future physicians to consider the patient’s perception on her or his condition [13,14].

The objective of this study was to gain a more profound understanding of how doctors categorize a condition, interpreting it as a disease or not, and what the lead of information they use in the process is. Another guiding question was what the purpose of such labelling is. The goal was to suggest to medical educators some theory-based pedagogies to create an integration of different definitions and points of view, helping their students to develop a more mature professional identity.

The research questions were

What is the information that doctors use to decide whether a condition is a disease?What are the perspectives from which doctors decide whether a condition is a disease or not?What purposes do doctors consider in labelling a condition as a disease?

## 2. Materials and Methods

We designed a survey based on a set of clinical vignettes and submitted it to a group of doctors and senior medical students, asking them to identify each condition as a disease or not and to motivate their decision.

As briefly discussed above, the definition of disease varies according to different ontologies and epistemologies. Hence, to acknowledge and manage this heterogeneity, we needed a flexible theoretical approach while analysing the data. For this reason, thematic analysis based on the critical realist approach was deemed the most appropriate.

This article is adherent to the Standard for Reporting Qualitative Research (SRQR) [15].

### 2.1. Theoretical Foundation

Thematic analysis is a technique that organizes and elaborates a set of data in detail. It is a means of finding, inductively analysing, and reporting patterns (themes) within the data [16]. By itself, thematic analysis can support different types of theoretical orientations in the interpretation of data. A critical realist approach was adopted to interpret the findings of this thematic analysis [17,18,19].

Critical realism holds that the universe, including the social world, exists. The basic ontology of critical realism holds that there are entities that can be material (an object), immaterial (a class), or both (an X-ray report, which is both information and a document). Entities have internal causal power and are related to each other in a world conceived as an open system [20]. The result of the interaction of entities and their power is not based on a linear cause-and-effect mechanism but is rather a pattern of events, that is, a tendency [17]. Critical realism admits that attempts to determine the nature of the world is fallible, but it is possible to gain a critical understanding of entities like the self, society, and its structures, and that one can develop theories specific to each domain. The task of a researcher is to develop a reliable explanation of events, tendencies, or outcomes, avoiding any rigid, context-based framework of cause-and-effect. Critical realism focuses on finding out what might work, how, and for whom, on a specific intervention, in a specific context. A critical realist account generally describes these variables in terms of contexts, mechanisms, and outcomes (CMOs). CMO configurations are used to explain why particular interventions have succeeded or failed and how the contexts and mechanisms affected these outcomes. Finally, critical realism embraces a view of the world as an open, stratified system with recurring looping effects, in which many factors other than the mechanisms considered intervene. A typical example of a looping effect is the relationship between a social classification and the behaviour of individuals in it. To the extent that they tend to adopt the expected behaviour, this reinforces the idea that the class is real and representative of that behaviour.

### 2.2. Reflexivity

All the authors were trained in qualitative research. During the preliminary discussion for the study design, no specific presumptions about the possible result emerged.

### 2.3. Context and Sampling Strategy

This study was conducted in a teaching hospital, submitting the questionnaire to academic physicians, senior medical students, and residents of different ages and genders. Some primary care physicians from the city area who served as primary care mentors were also involved. The sampling was purposive and aimed to form a heterogeneous group of respondents with responses as diverse and rich as possible. This was a qualitative study, and qualitative research has no presumption of generalisability but aims to bring out some elements of a complex construct. We thought that the most fundamental elements of heterogeneity were sex, age, and field of medical expertise, and the selection of subjects were balanced according to these characteristics.

The authors personally approached the subjects to join this study, to write their answers, and to return the questionnaire in a post box within one week, to grant anonymity. The questionnaires were progressively numbered.

### 2.4. Data Collection Methods

The questionnaire was designed according to the model of representation of disease proposed by Tresker [21], who developed a typology of disease formed from three factors: presence of symptoms, presence of a dysfunction (intended in Boorse’s term as an objective alteration of physiology [2]), and the concordance with diagnostic criteria. The intersection of these factors created eight categories. This proposal had the advantage of encompassing the three approaches to the definition of disease mentioned in the Introduction.

Sixteen clinical vignettes were designed, two for each category. The vignettes were assessed for face validity by a panel of three experienced physicians, who evaluated their correctness and whether they were representative of the eight Tresker’s categories. The respondents were shown only the vignettes and not how each situation was categorized according to Tresker [21]. The responders were asked if they considered each situation a disease or not and to motivate their choice with a free text.

Table 1 shows the eight categories and an example vignette for each.

### 2.5. Data Processing and Analysis, Trustworthiness

The answers to the questionnaire were transcribed in a word processor, and each medical author had a copy of the transcript. All medical authors read the transcripts and analysed them according to thematic analysis. More specifically, each author read the transcript twice, then identified the units of meaning (words or phrases that conveyed meaning of interest to the research questions) and coded them, applying a word or short phrase that expressed the meaning. Periodically, during the analysis process, the team met and compared the choices of units and codes, discussing until a reaching a common agreement. These discussions were the main mechanism for strengthening trustworthiness.

After analysing about half of the texts, a common codebook was established and used to code the rest of the texts. When it was felt that a new code was needed, it was noted, discussed, and added to the codebook. When no new codes emerged, saturation was declared.

Finally, F.C., L.G. and D.P. inductively collected the codes into themes and sub-themes, separately and then with joint discussion to resolve any disagreements. These results were interpreted according to a critical realist approach, highlighting the CMO elements.

### 2.6. Ethical Issues

The survey was anonymized, and all the participants were informed of the method and goal and gave their consent to join this study. The questionnaire was anonymous, no sensitive data were permanently recorded, and this study did not imply any direct intervention with the subjects; hence, according to Italian law and regulations, an IRB ethical authorization was not needed. This is compliant with the stipulations of European law: The General Data Protection Regulation (GDPR) (EU) 2016/679.

## 3. Results

Nineteen subjects returned the questionnaire. Table 2 reports their characteristics.

Saturation was achieved after 14 questionnaires, but all the returned questionnaires were analysed.

Thematic analysis showed four main themes, with ten sub-themes. The themes were the temporal dimension of a disease, reification of disease, disease as an existential condition, and disease as a motivation to action. Table 3 summarizes the themes and sub-themes, which are described in the following paragraphs with some relevant quotes from the questionnaires. Every quote has been labelled with the number of the respondent, with the letter identifying the vignettes categorised according to Tresker’s typology(Table 1) and with Y/N according to the interpretation that the respondent gave of the vignette. The original data were in Italian, the reported quotes have been translated into English.

### 3.1. Correspondence of the Themes with the Research Questions

The four emerged themes, with their sub-themes, allowed us to answer the research questions of this study. As to the first research question, we found that two classes of information (objective and subjective data) influenced judgement, with the mechanisms described more in detail in the first three themes. The third and fourth themes showed that two perspectives emerged (patient’s perspective, doctor’s aims). Finally, the only quoted purpose for labelling a condition as a disease was motivation to action.

### 3.2. A Disease Is an Event That Occurs over Time

Direct or indirect temporal expressions were present in the motivation for the interpretation, related to the clinical nature of the problem (acute vs. chronic) and to its possible evolutions. Temporal nature was mainly used to characterize the problem, whether it was considered a disease or not (an acute event): “A disease was not diagnosed as a cause of the acute event. He had an undiagnosed acute event, he was healthy” (4, G, N); “Although not recognised in its aetiology, an acute disease brought the man to exitus. He had an acute disease.” (5, G, Y).

The analysis of data also highlighted the concept of evolution over time of a health-related condition. At this proposal, one of the most illuminating quotes was “We are dynamic beings; an alteration for a short period of time cannot be called a disease.” (9, B, N). To the same extent, another quote was “The persistence of a discomfort is a reason for classifying a condition as a disease” (1, F, Y).

Finally, in the following two quotations, the risk condition was interpreted as either an element of disease or an element of non-disease, depending on the attributed evolutionary potentiality: “The woman suffers from a precancerous lesion, caused by a chronic infection. It is an objective condition with a possible degeneration” (4, B, Y); “The condition [of cervical intraepithelial neoplasia] may regress or remain stable over time” (13, B, N).

### 3.3. Reification of Disease: A Disease Is a Knowable Object of the World

This is the broadest theme, as it collected all mentions of an objective biochemical, pathological, or functional alteration. Instances of causal reasoning have also been included in this theme, as well as the recognition of known criteria for a specific diagnosis.

Examples of the first sub-themes were “This woman is not ill, at present she has not any sign or symptoms, nor evidence of amyloid plaques” (3, A, N); “The patient has a cervical intraepithelial neoplasia, demonstrated with a colposcopy” (3, B, Y). Nevertheless, this condition was not recognised by another respondent as a disease, because “There is no consequence on the integrity of the self” (7, B, N). Instances of the importance of a cause-and-effect sequence were present, such as “It seems to be a sudden acute episode, without any specific cause, in a man that was otherwise healthy” (11, G, N). Nevertheless, another respondent gave a different interpretation for the same vignette, implying a cause-and-effect sequence: “The postmortem assessment documented a previous disease as the cause of the acute event” (14, G, Y). Also in vignette B, the condition was considered as a disease by the respondent whose choice was motivated as follows: “The patient has a viral infection, which caused a dysplasia” (14, B, Y).

Finally, a disease was recognised when diagnostic criteria were met, as in “This patient meets the Rome criteria for an irritable bowel syndrome” (14, F, Y).

### 3.4. A Disease Is an Existential Condition in a Person’s Life

Many respondents explicitly adopted the patient’s point of view in deciding if a condition was a disease or not. Subtle distinctions were made, which were classified as different sub-themes, such as the perception of a general disorder of a person’s life, a specific disorder or set of symptoms declared by the patient, or a sudden change in context or setting in the patient’s life, even without any symptoms.

The same respondent reacted in a different way to vignettes A and C, using the concept of compromission of the quality of life: “At present, the condition did not alter in any way the quality of life of the woman, and she is not necessarily going to develop the disease” (11, A, N) and “The BRCA-2 mutation, beyond being an organic, genetic disfunction, compromised the overall psycho-physical state of the woman” (11, C, Y). In the same case, another respondent was even more radical, stating that “The woman is ill, because the awareness of the mutation changed her way of looking at herself and originated a decision” (6, C, Y).

In vignette E, considerations more focused on symptoms motivated the interpretation of a condition as a disease: “A condition of recurrent abdominal pain implies an altered health status and impairs a person’s organic and psychological function.” (8, E, Y). A “persistence of the discomfort” (7, E, Y) in itself is a motivation in considering a condition as a disease.

Finally, the concept of setting and the way health information was communicated to patients were cited as elements that may or may not make a person consider that condition a disease. For example, respondent 5 wrote that “the perception of risk and probability is strongly influenced by the way in which it is communicated to the patient. This could be a distressing element in a person’s life” (5, C, Y).

### 3.5. A Disease Is a Motivation to Action for a Physician or a Patient

Some choices were motivated by the need for the physician or the patient to do something. The two classes of actions were clinical actions like treatment or preventive actions like follow up or even surgery. Examples of these sub-themes were “The woman has a precancerous condition…objectively treatable” (4, B, Y), but the same respondent paradoxically concluded that in the apparently similar case C (a woman with the BRCA2 mutation), the woman was not ill, because “she underwent surgery to prevent the likely onset of cancers, but she is still healthy” (4, C, N). Hence, if the medical action was treatment, the condition was interpreted as a disease; if the action was the prevention of a risk, the condition was not interpreted as a disease. Another respondent wrote, “A CIN is present, that does not require a treatment” (9, B, N). Despite the different considerations about the treatability of a CIN, the two respondents agree on the fact that the need for treatment is a landmark for the presence of a disease.

A similar paradox was present in the case of a 59-year-old man who was asymptomatic and had all the findings of a metabolic syndrome (type D). One respondent identified the man as having the criteria for a diagnosis but not illness: “This man has all the risk factors for a metabolic syndrome, but he is not actually ill. He needs to change his dietary habits and follow up.” (2, D, N).

## 4. Discussion

This study answered the research questions and showed that physicians used a fairly broad set of information to decide whether a condition is a disease or not, but they used this information very heterogeneously. The same information would lead two physicians to different conclusions, and sometimes the same physician would interpret two similar situations in a different way. This finding is consistent with the result of Hoffman’s study [10], which did not support the presence of a common prototypical concept of disease. Our result is also consistent with the classic distinction between the concepts of “disease” and “illness” [9], which respectively represent an objective, physician-centred or subjective, patient-centred definition of a health condition.

Further, this study showed that this heterogeneous group of physicians, in reasoning about the nature of a condition, considered the temporal dimension, both as a clinical aspect and as a process in time that begins before or follows something else, not necessarily in a cause-and-effect relationship but as a risk or a probabilistic evolution [22].

Finally, in considering the question on what purposes doctors consider in labelling a condition as a disease, the physicians seemed to take a pragmatic approach to interpret a condition, which was not based on fixed-principle reasoning. This conclusion is consistent with Hesslow’s doubt about the need for a formal definition of disease [7].

### 4.1. CMO Interpretation

Interpreting these results in terms of CMO configuration, the outcome is the interpretation of a situation as a disease or not, and the mechanisms are the four classes of information thematized as temporal dimension, reification, existential dimension, and call to action. A fundamental tenet of critical realism is that the context can change the likelihood of a mechanism to produce an outcome. Our results showed a very heterogeneous relationship between mechanisms and the type of outcome; hence, our hypothesis is that the context perceived by the respondents has been the source of this heterogeneity.

A relevant element of context is the doctor–patient relationship. A characteristic and limitation inevitable in this study, like Hoffman’s [10], was that physicians were surveyed about a context-free situation, with a series of written vignettes or a quantitative survey. There was neither information nor lived experience of a real situation in which a doctor and a patient meet. Many studies have shown that the physician–patient relationship affects the physician’s interpretation of the patient’s condition. Most studies have investigated the effect of relational context on clinical reasoning, understood as the general cognitive mechanism by which a physician makes a diagnosis [23,24], while Krimmel-Morrison et al. [25] have delved into the effect that a relationship has on physicians’ approaches to diagnosis and management and to building and preserving relationships with patients. The themes that emerged were “Adapting clinical reasoning through the lens of patients’ experiences”, “Managing tensions between clinical reasoning and relationships”, and “Clinical reasoning within patients’ other influential relationships”, such as family, friends, and caregivers. It is likely that the respondents in our study implicitly and unconsciously adopted a fictitious context, and this may explain contradictions not only among different respondents, but sometimes even within the responses of the same individual: the same mechanism, in a different imagined context, generated a different outcome.

### 4.2. Other Possible Interpretations and Limitations

Other possible interpretations can be considered. Livingstone-Banks [6] argued that, for any group of people, there are multiple ways of grouping these people according to their health states, and thus multiple candidate medical taxonomies exist. Therefore, the heterogeneity in the interpretation of a health condition could be based on each physician’s epistemological premises, which can be more prone to a naturalist, normative, or pragmatic stance. This could explain the variability in interpretation among physicians but less so the difference in judgment of the same physician when faced with similar situations.

An interesting suggestion comes from Hofmann [26], who identified six types of disease based on knowledge of the mechanisms of disease (epistemic); the phenomena that define disease (ontological); what we can do about it (pragmatic); what we define as disease (conceptual); what we define as bad (ethical); and what we consider bad (aesthetic). This model suggests that it might be interesting to explore how the “feeling” of something wrong, bad, or unpleasant influences a physician’s interpretation of a health condition.

This study had some limitations due to the nature of qualitative research. Our conclusions are not generalisable and valid for all physicians. This study showed that the mechanisms we described are at play in our respondents and that they are real. Other quantitative studies should determine in standardized samples how widespread the different attitudes are, while qualitative studies should explore whether the hypothesis we advanced about the doctor–patient relationship as a context for interpreting a disease is real.

Other limitations of this study were that, although varied in age, sex, professional seniority, and field of specialization, most of the respondents were from the same institution, and all of them were Italian. It is possible that a different group could show other mechanisms and tendencies [27].

### 4.3. General and Educational Implications

One of the main reasons physicians need a clear understanding of what a disease is that pharmaceutical and technological advances are expanding the range of situations in which a physician can intervene. However, broadening the definition of a risk factor or disease can produce overdiagnosis or overtreatment as a dangerous side effect [28]. This is also relevant from the perspective of patient education as a basis for informing public campaigns. It must be clear that there is no value-free definition of disease [29]. In their discussion of three examples of diseases (polycystic ovary syndrome, kidney chronic failure, and myocardial infarction) the authors concluded that “we need theories of disease that are transparent about the values at stake, and that can be used to prevent problems that arise from indistinct boundaries.” ([29], p. 352).

Hence, a first commitment for medical educators must be the design of training activities in which students can be challenged with various case-based and living situations and reflect on the values at stake. Both the Tresker’s framework [21] that was used in this study and the framework proposed by Hofmann and Eriksen [30] can be used. This last framework builds on the triad of disease–illness–sickness, allowing a guided reflection on the different mechanisms in action in different contexts in real or simulated observed cases, or on-paper case-based learning activities.

Another useful theoretical framework was proposed by Koufidis et al. [31], who in their scoping review identified three main conceptualizations of clinical reasoning: “reasoning as a cognitive activity”, “reasoning as a contextually situated activity”, and “reasoning as a socially mediated activity”. These perspectives can broaden students’ understanding of the reasoning process—which leads not only to a diagnosis but also to an interpretation of the patient’s lived experience of discomfort—and on the consequences of each physician’s choice.

Finally, the results of the present study have shown that the method used (vignettes and related questions) can encourage reflection and make some implicit assumptions more explicit, leading the learner to have more elements to develop different interpretations of a clinical condition. This can be more effective when comparing several similar situations described in the vignettes or observed in real patients, because this can make the effect of context more apparent.

## 5. Conclusions

From a critical realist and systemic perspective, there is no reason to be forced to choose between a naturalist or social and normative definition of the concept of disease. If one considers the person as an open adaptive system, consisting of a body, a mind, a spirituality, a personal history, and a broad set of relationships, each subsystem is important and deserves the utmost attention. A genetic mutation that could give rise to an aggressive cancer in the future or a biochemical alteration in an otherwise asymptomatic man, a medically unexplained symptom in a young woman or a social and economic situation that exceeds the adaptive capacity of an elderly person or an entire social group are all mechanisms that can produce an alteration in health as an outcome. This study has added some insights to the soundness of a critical realist position, which cannot be regarded as a pragmatic, hybrid approach, but a theoretically grounded one.

Future research should test and assess the validity of the proposed educational activities.

## Figures and Tables

**Table 1 healthcare-12-01228-t001:** The eight categories and examples of clinical vignettes. In bold is the information that is consistent with the factors according to Tresker’s typology [21]. The elements that characterize each situation are also in bold.

Examples of Clinical Vignettes, Classified According to Tresker’s Typology
*A. Asymptomatic person, without dysfunction, and not meeting diagnostic criteria for a disease.*
A healthy 35-year-old woman, family history of early Alzheimer’s (mother, with onset at age 50). Genetic testing suggested a predisposition for disease development. Periodic neuroimaging examinations were recommended so as to highlight presymptomatic beta-amyloid accumulation.
*B. Asymptomatic person, without dysfunction, but meeting diagnostic criteria for a disease.*
A healthy 25-year-old woman. She had periodic Pap tests, the last of which demonstrated “atypia of uncertain significance.” The viral DNA search was positive, and a colposcopy demonstrated the presence of a **cervical intraepithelial neoplasia (CIN)**, for which no therapeutic indication was placed but only follow up.
*C. Asymptomatic person, with dysfunction, not meeting diagnostic criteria for a disease.*
A 41-year-old woman with a maternal history of breast cancer based on the BRCA2 mutation. After genetic testing, the woman was also found to have a **BRCA2 mutation**. PE, breast ultrasound, mammogram, MRI, gynaecological examination, and trans-vaginal ultrasound were negative; CA125 was normal. After 18 months of follow up, the woman agreed to undergo bilateral mastectomy and hystero-annexectomy.
*D. Asymptomatic person, with dysfunction, meeting diagnostic criteria for a disease.*
A 25-year-old woman, a nurse on regular duty, was used to act out self-injurious behaviours (scratches on legs) in private, which she hid under her uniform from colleagues and patients A psychiatrist matched the criteria for a diagnosis of **schizoid syndrome**. Except for some mild **mood swings**, which she controls pharmacologically, at present she has no problems at work.
*E. Symptomatic person, without dysfunction, and not meeting diagnostic criteria for a disease.*
A 50-year-old man with **recurrent abdominal pain** in the right lower quadrant started about a year ago and no change in bowel habits. Appendectomy 10 years ago. PE negative for parietal hernias, WBC normal, CT scan of abdomen, and pelvis negative for bowel and renal changes, no lymphadenopathy.
*F. Symptomatic person, without dysfunction, meeting diagnostic criteria for a disease.*
A 25-year-old woman with **recurrent abdominal pain and increased bowel movement with soft stools in the past 6 weeks**. No weight loss; PE not significant; negative faecal test for pathogenic bacteria, parasites, test for celiac disease, and calprotectin. Negative abdominal ultrasound. Despite a diet correction, the disorder persists. The woman reports having a series of job interviews in progress.
*G. Symptomatic person, with dysfunction, not meeting diagnostic criteria for a disease.*
A 70-year-old man in good health complains of an sudden onset of acute, diffuse postprandial abdominal pain.In the ER, the man is **agitated**, **confused** at times; his **abdomen is diffusely tender**, but without defensive contractures. **Blood pressure 220/110**. Despite treatment initiated, within hours, exitus occurs. Autopsy documents extensive bowel infarction that had not been suspected or diagnosed.
*H. Symptomatic person, with dysfunction, meeting diagnostic criteria for a disease.*
84-year-old man with a history of **COPD**. The man is **confused, temperature 38.7 °C, dyspnoea, productive cough, saturimetry = 88%**.

**Table 2 healthcare-12-01228-t002:** Description of the respondents.

Characteristic	N.
Sex	Men: 9 Women: 10
Age	26÷30 years old: 8 >30 years old: 11
Role and speciality	Students: 4 Surgeons: 5 Internal medicine: 8 General practitioners: 2

**Table 3 healthcare-12-01228-t003:** List of themes and sub-themes identified with the thematic analysis.

Theme	Sub-Themes
Temporal dimension of a disease	Temporal nature: acute vs. chronic conditions Evolution over time of a condition
Reification of disease	Disease as an objective biochemical, pathological, or functional alteration Disease as an outcome of a cause-and-effect mechanism Disease as a condition based on diagnostic criteria
Disease as an existential condition	Disease as a general disorder of a person’s life Disease as a specific disorder or set of symptoms Disease as a context or setting
Disease as a motivation to action	Motivation to clinical action (treatment) Motivation to preventive action (follow up, change in life habits)

## Data Availability

Data are in part contained within the article. Full data are available on request from the corresponding author.

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
