# Peer review of "What Is a Disease for Doctors? A Realist Thematic Qualitative Analysis of the Interpretation of Clinical Vignettes"

_healthcare, 2024, doi:10.3390/healthcare12121228_

Round 1

Reviewer 1 Report

Comments and Suggestions for Authors

This paper was a qualitative exploration of physicians' perspectives and thoughts on the concept of disease and the scope of defining it, with implications for medical education. However, the most concerning aspect of the research is that it was not IRB-cleared, even though it was a human subjects survey.  I am not familiar with IRB regulations in Italy, but as far as I know, all kinds of human subjects research worldwide, even if it is anonymous and non-invasive, must go through an IRB. Even though the ethical issues would be resolved, I think the paper will need major revisions before it can be published.

First, the title is ambiguous. The subtitle mixes the methodology and the purpose of the study in parallel with other elements.

Second, the abstract and key words need to be rewritten. The research methods and facts should be clearly presented. For example, you should explain how and when you collected data, from how many people, and how you analyzed it. The conclusion should be based on the findings and should be complete and in line with the purpose of the research. Keywords should include terms and the country name that represent the main findings of the paper.

Third, you should explain your findings in a way that meets your research objectives. RQ1 is to explore the criteria and information for judging a disease as a disease, but the results of the study do not match the specific explanation. The findings are only explored in a descriptive manner, which does not reveal the "thick description" that the original authors intended, and it is not clear what the authors are trying to convey in the paper.

Fourth, the methodology needs to be completely revised. The sampling methodology is not clear and it is not clear whether it is a purposive or snowball sample. The researchers claim to have considered age and gender, but the characteristics of the participants are not shown and it is not clear how they were selected. The recruitment sites are very diverse, both in training hospitals and in primary healthcare centers throughout cities of Italy, and it is questionable whether it is meaningful to present the results in the same group, given the different severity of illnesses treated by physicians in tertiary hospitals and primary healthcare centers. Participant characteristics such as length of clinical experience may also affect the results.

Finally, the study participants were very heterogeneous, which reduces the validity and reliability of the results. As the authors note in the abstract, the results are very mixed, which I believe is due to the lack of a purposive sampling design to focus on answering the research questions. I recommend that the study be conducted with full IRB approval and a redesign of the study sample.

Comments on the Quality of English Language

The authors need to polish their English a bit to make it easier for readers to understand.

Author Response

Dear Editor and Reviewers,

thank you for the effort and the time you devoted to reviewing our manuscript.  

We would like to clarify one main point of concern right away: the IRB approval. In Italy the topic of ethical approval of research is under two different jurisdictions. All clinical, pharmacological and genetic research is ruled by a Ministry of Health decree issued on Feb. 8, 2013, which requires approval by a local or regional ethics committee.
All other types of non-clinical research are under the national 196/2003 act, also named Privacy Code. The act rules all types of management and use of personal data, whatever the goal is. Protection must be particularly strong for sensitive data, that can disclose opinions, believes, attitudes and choices.

The study reported in the manuscript falls under the 196/2003 act, because it is non-clinical research, and is compliant with the act because:

- data are fully anonymised. Anonymity was ensured by the way the questionnaires were returned, using a mailbox.

- the respondents had full information about the method and the goal of the study and voluntarily joined it.

We knew who had joined the study, but it was not possible to pair their response with them. If we had asked the hospital or university IRB for approval, the request would not have been considered.

In the following paragraphs, we answer to the reviewers’ requests

Reviewer 1 report

First, the title is ambiguous. The subtitle mixes the methodology and the purpose of the study in parallel with other elements.

We agree and have changed the title.

Second, the abstract and key words need to be rewritten. The research methods and facts should be clearly presented. For example, you should explain how and when you collected data, from how many people, and how you analyzed it. The conclusion should be based on the findings and should be complete and in line with the purpose of the research. Keywords should include terms and the country name that represent the main findings of the paper.

We have rewritten the Abstract and key words according to these suggestions. More in particular:

  • we added some more details about the methodology
  • we rephrased the final sentence, to highlight that the conclusions were based on the four resulted themes
  • we added four more key words

Third, you should explain your findings in a way that meets your research objectives. RQ1 is to explore the criteria and information for judging a disease as a disease, but the results of the study do not match the specific explanation. The findings are only explored in a descriptive manner, which does not reveal the "thick description" that the original authors intended, and it is not clear what the authors are trying to convey in the paper.

The recruitment sites are very diverse, both in training hospitals and in primary healthcare centers throughout cities of Italy, and it is questionable whether it is meaningful to present the results in the same group, given the different severity of illnesses treated by physicians in tertiary hospitals and primary healthcare centers. Participant characteristics such as length of clinical experience may also affect the results. Finally, the study participants were very heterogeneous, which reduces the validity and reliability of the results. As the authors note in the abstract, the results are very mixed, which I believe is due to the lack of a purposive sampling design to focus on answering the research questions.

We are not sure we got the point in these two comments, that we put together.
Qualitative analysis aims to describe a phenomenon in its elements and reveals what exists. This is why results are “only” description and are derived from a rigorous method of analysis. Our study is based on a qualitative method (thematic analysis) and on a theoretical perspective (critical realism).
Validity and reliability are concepts related to quantitative research and, more in particular, to an instrument to “measure” a mental construct
or a biological value. The goal of quantitative research is to answer to questions such as “how many/frequent?”, “what is the correlation/effect?” and so on. The dimension and characteristics of the sample are very important for the generalisability of the results.

In qualitative research the more heterogeneous the sample, the better it is. There is not any assumption of generalisability, the aim is to show that “something” exists in the world. In this study, we looked for “mechanisms”, in the terminology of critical realism, through which a doctor decides if a condition should be considered a disease or not. We found that two classes of information (objective and subjective data) influenced the judgement (RQ1), that two perspectives emerged (patient’s perspective, doctor’s aims- RQ2) and that the only quoted purpose of labelling a condition as a disease was motivation to action (RQ3)

We are not contesting the reviewer’s judgement. On the contrary, we are grateful for her/his comments, which made us understand that the text was unclear in explaining the nature and the results of the study.
We edited the Results and the Discussion, adding some of what we wrote above, for the benefit of the readers. We hope you and the editor will approve our effort of clarity.

I recommend that the study be conducted with full IRB approval and a redesign of the study sample.

We preliminarily discussed the IRB issue

Fourth, the methodology needs to be completely revised. The sampling methodology is not clear and it is not clear whether it is a purposive or snowball sample. The researchers claim to have considered age and gender, but the characteristics of the participants are not shown and it is not clear how they were selected.

We explained in Methods that the sampling was purposive and added in Results the characteristics of the sample study group.

Reviewer 2 Report

Comments and Suggestions for Authors

Thank you for your paper i enjoyed reading qualitative research in medicine, and highly encourage it. Overall the research was conducted and written very well, lacking for me is a passion in the paper, a more compelling reason why this is so important. Maybe add one or two paragraphs that address this. Thank you.

Comments on the Quality of English Language

good, only saw one main mistake, line 89 , can you find it?

Author Response

Thank you.

We found the missing "s" and corrected

Reviewer 3 Report

Comments and Suggestions for Authors

Dear authors,

thank you for your work in this field. I have a few questions about your study:

a) the title, apart from the capitalized one, should be revised. In addition to the implications for medical education, there are implications for research and society in general;

b) I think the text on line 78 should be moved to point 2 of Materials and methods;

c) I don't think it's necessary to list the competences of researchers in point 2.2 either. At the end of the article, line 350, each person's contribution is listed;

d) Line 127 et seq. states that there is no need for an ethics committee approval, is this the case for all qualitative and quantitative studies involving human volunteers? Do hospitals allow inquiries to be carried out on professionals who have not passed through ethics committees? 

e) I also didn't find any form for informed consent for the study volunteers in the text.

f) Was the design of the questionnaire pre-tested?

g) The sample includes doctors and medical students, how can conclusions be drawn for the whole medical profession if they include staff in training? Don't you bias the findings?

h) The choice of sample is not indicated. Was it random? Was it convenient? Were there any exclusion criteria? 

i) The sample is not characterized (age, gender, length of service, care unit, specialty, etc.). This point is important, not least for the discussion of the results;

j) Data saturation occurred with 14 questionnaires, but 19 were analyzed, right? This doubt arises because, as I said earlier, there is no characterization of the sample;

l) I think the discussion of the results could be more in-depth. Although there are few studies, the authors can raise possible hypotheses that justify the results. 

m) I don't see any research implications for your study either. You need to look into this further for future research.

Author Response

Dear Editor and Reviewers,

thank you for the effort and the time you devoted to reviewing our manuscript.  

We would like to clarify one main point of concern right away: the IRB approval. In Italy the topic of ethical approval of research is under two different jurisdictions. All clinical, pharmacological and genetic research is ruled by a Ministry of Health decree issued on Feb. 8, 2013, which requires approval by a local or regional ethics committee.
All other types of non-clinical research are under the national 196/2003 act, also named Privacy Code. The act rules all types of management and use of personal data, whatever the goal is. Protection must be particularly strong for sensitive data, that can disclose opinions, believes, attitudes and choices.

The study reported in the manuscript falls under the 196/2003 act, because it is non-clinical research, and is compliant with the act because:

- data are fully anonymised. Anonymity was ensured by the way the questionnaires were returned, using a mailbox.

- the respondents had full information about the method and the goal of the study and voluntarily joined it.

We knew who had joined the study, but it was not possible to pair their response with them. If we had asked the hospital or university IRB for approval, the request would not have been considered.

In the following paragraphs, we answer to the reviewers’ requests

Reviewer 3 report

  1. a) the title, apart from the capitalized one, should be revised. In addition to the implications for medical education, there are implications for research and society in general;

We changed the title

  1. b) I think the text on line 78 should be moved to point 2 of Materials and methods;

We moved the line

  1. c) I don't think it's necessary to list the competences of researchers in point 2.2 either. At the end of the article, line 350, each person's contribution is listed;

Point 2.2 is a requirement of SRQR standard, but we agree that in this study it may be omitted. We erased the first part of the section

  1. d) Line 127 et seq. states that there is no need for an ethics committee approval, is this the case for all qualitative and quantitative studies involving human volunteers? Do hospitals allow inquiries to be carried out on professionals who have not passed through ethics committees?

This issue has been discussed at the beginning of this cover letter 

  1. e) I also didn't find any form for informed consent for the study volunteers in the text.

Thank you, we forgot this point and added a sentence

  1. f) Was the design of the questionnaire pre-tested?

Good point, thank you. Yes, it was. We added a sentence

  1. g) The sample includes doctors and medical students, how can conclusions be drawn for the whole medical profession if they include staff in training? Don't you bias the findings?

Qualitative research is not meant to draw generalizable conclusions, but only to show some of the elements of the inner structure of a phenomenon. For the benefit of the readers who may be not familiar with qualitative research, we explained this point with more details in the Discussion

  1. h) The choice of sample is not indicated. Was it random? Was it convenient? Were there any exclusion criteria?

We added a sentence in Methods, to provide this information

  1. i) The sample is not characterized (age, gender, length of service, care unit, specialty, etc.). This point is important, not least for the discussion of the results;

We added a description of the characteristics of the participants in the Results

  1. j) Data saturation occurred with 14 questionnaires, but 19 were analyzed, right? This doubt arises because, as I said earlier, there is no characterization of the sample;

Correct. After 14 questionnaires, no more new codes emerged. Nevertheless, we went on in analysing all the filled questionnaires, to increase the trustworthiness of the analysis.

  1. l) I think the discussion of the results could be more in-depth. Although there are few studies, the authors can raise possible hypotheses that justify the results.

We expanded the Discussion with two more possible interpretations

  1. m) I don't see any research implications for your study either. You need to look into this further for future research.

Again, a good suggestion, thank you. We added some lines in the Discussion

Reviewer 4 Report

Comments and Suggestions for Authors

This paper brings to attention of the readerships a very important and timely topic, that of, first, bringing attention to how 'concepts and words' are often taken at face-value, with shared conception and definitions among various individuals, groups or professionals, but that in reality, there can be important differences, and that these can have an impact on practices and services -- in the context of the article, care and medical practices.

I greatly enjoyed reading this study! The paper is clear, well writing, scientifically valid; the research is presented appropriately, with sufficient and appropriate details to contextualize the study, explain and justify the methods, present and discuss the results, explain and justify the relevance of conducting the study, and connecting with existing literature as well as timeliness of the study.

It is all too rare, too, that such study, in my view and knowledge, is featured in journals, so I was very satisfied to read it, discover it, evaluate it, and now, to recommend its publication - and I thank the authors for this work.

Gaining an understanding from empirical research of how concepts are defined, and influence possibly practices, is of great importance, as well as, I would say, 'key' in improving practices, increasing the respect of patients and families and their lived experiences, improving the conditions of care as well and supporting well carers and health professionals, who are often placed in situations of uncertainty, the unknown, doubt -- and making evident how the 'real/physical world' and the 'social/historical/political/cultural worlds' influence one another, and interact, can but only support better carers in caring for others (providing care, treatments, etc.)

I think this piece is ready for publication and see no need to recommend any additional work on the parts of authors, minor or major. Thank you. 

Author Response

We wish to thank you for your very encouraging comments on our manuscript.
Our first idea was to submit to a specialized journal of philosophy of medicine, but we are clinicians and wanted to communicate these ideas to colleagues working daily on the field.

Thank you again

Round 2

Reviewer 1 Report

Comments and Suggestions for Authors

First of all, I am pleased that the authors have taken the effort to improve the quality of their paper for publication in the journal Healthcare. The revised manuscript presents the perspective and results of the study more clearly. I would like to suggest the authors to complete a second round of revisions to improve the paper.

1. I recommend adding a table describing the characteristics of the study participants. 

As the authors state, qualitative research does not require a large sample size because it is not intended for generalization, but in the case of purposive sampling, there should be a sufficient explanation of why the authors wanted to hear from different groups of doctors, and the characteristics of the participants should be presented in a way that reveals the authors' intentions. 

2. Table 1 is not in the format used in the academic paper. Please insert a title line to the table to make it more complete. 

3. The contents of subheading 3.5 only has a title and a one-line description, please adjust the length to be similar to other subtopics.

4. Please consider the keyword count limit in the abstract and select the keywords that are essential. For example, consider deleting illness and sickness from the abstract as they are not directly discussed in the abstract.

Author Response

First of all, I am pleased that the authors have taken the effort to improve the quality of their paper for publication in the journal Healthcare. The revised manuscript presents the perspective and results of the study more clearly.

We are happy to read this, thank you

I would like to suggest the authors to complete a second round of revisions to improve the paper.

  1. I recommend adding a table describing the characteristics of the study participants.

We added a table

As the authors state, qualitative research does not require a large sample size because it is not intended for generalization, but in the case of purposive sampling, there should be a sufficient explanation of why the authors wanted to hear from different groups of doctors, and the characteristics of the participants should be presented in a way that reveals the authors' intentions.

We added a sentence (125-127): “We thought that the most fundamental elements of heterogeneity were sex, age and field of medical expertise and be balanced the selections of subjects according to these characteristics”

  1. Table 1 is not in the format used in the academic paper. Please insert a title line to the table to make it more complete.

We added a line

  1. The contents of subheading 3.5 only has a title and a one-line description, please adjust the length to be similar to other subtopics.

Perhaps you refer to subheading 2.2. Reflexivity is one of the points mandated by the SRQR standard. It has been shortened according to a suggestion of reviewer 3. We acknowledge that that sub.section seems “strange”, but we must balance the different reviewers’ suggestions.

  1. Please consider the keyword count limit in the abstract and select the keywords that are essential. For example, consider deleting illness and sickness from the abstract as they are not directly discussed in the abstract.

We deleted the two keywords

Reviewer 3 Report

Comments and Suggestions for Authors

I think the authors have made improvements to the article following suggestions. However, I believe that they should include them in their article, as they have not been endorsed by an ethics committee, at least they have respected the stipulations of European law:  The General Data Protection Regulation (GDPR) Regulation (EU) 2016/679

Author Response

I think the authors have made improvements to the article following suggestions.

We are happy to read this, thank you

However, I believe that they should include them in their article, as they have not been endorsed by an ethics committee, at least they have respected the stipulations of European law:  The General Data Protection Regulation (GDPR) Regulation (EU) 2016/679

we added a sentence (lines 170.171): “This is compliant with the stipulations of European law:  The General Data Protection Regulation (GDPR) Regulation (EU) 2016/679.”